# Integrated Metabolomics and Lipidomics Analysis Reveals Lipid Metabolic Disorder in NCM460 Cells Caused by Aflatoxin B1 and Aflatoxin M1 Alone and in Combination

**DOI:** 10.3390/toxins15040255

**Published:** 2023-03-31

**Authors:** Xue Yang, Xue Li, Yanan Gao, Jiaqi Wang, Nan Zheng

**Affiliations:** 1Key Laboratory of Quality & Safety Control for Milk and Dairy Products of Ministry of Agriculture and Rural Affairs, Institute of Animal Sciences, Chinese Academy of Agricultural Sciences, Beijing 100193, China; 2Laboratory of Quality and Safety Risk Assessment for Dairy Products of Ministry of Agriculture and Rural Affairs, Institute of Animal Sciences, Chinese Academy of Agricultural Sciences, Beijing 100193, China; 3Milk and Milk Products Inspection Center of Ministry of Agriculture and Rural Affairs, Institute of Animal Sciences, Chinese Academy of Agricultural Sciences, Beijing 100193, China; 4State Key Laboratory of Animal Nutrition, Institute of Animal Sciences, Chinese Academy of Agricultural Sciences, Beijing 100193, China; 5Research and Development Institute, Heilongjiang Feihe Dairy Co., Ltd., Qiqihar 161000, China

**Keywords:** aflatoxin B1, aflatoxin M1, metabolomics, lipidomics, NCM460, enterotoxicity

## Abstract

Aflatoxin B1 (AFB1) and aflatoxin M1 (AFM1) are universally found as environmental pollutants. AFB1 and AFM1 are group 1 human carcinogens. Previous sufficient toxicological data show that they pose a health risk. The intestine is vital for resistance to foreign pollutants. The enterotoxic mechanisms of AFB1 and AFM1 have not been clarified at the metabolism levels. In the present study, cytotoxicity evaluations of AFB1 and AFM1 were conducted in NCM 460 cells by obtaining their half-maximal inhibitory concentration (IC50). The toxic effects of 2.5 μM AFB1 and AFM1 were determined by comprehensive metabolomics and lipidomics analyses on NCM460 cells. A combination of AFB1 and AFM1 induced more extensive metabolic disturbances in NCM460 cells than either aflatoxin alone. AFB1 exerted a greater effect in the combination group. Metabolomics pathway analysis showed that glycerophospholipid metabolism, fatty acid degradation, and propanoate metabolism were dominant pathways that were interfered with by AFB1, AFM1, and AFB1+AFM1. Those results suggest that attention should be paid to lipid metabolism after AFB1 and AFM1 exposure. Further, lipidomics was used to explore the fluctuation of AFB1 and AFM1 in lipid metabolism. The 34 specific lipids that were differentially induced by AFB1 were mainly attributed to 14 species, of which cardiolipin (CL) and triacylglycerol (TAG) accounted for 41%. AFM1 mainly affected CL and phosphatidylglycerol, approximately 70% based on 11 specific lipids, while 30 specific lipids were found in AFB1+AFM1, mainly reflected in TAG up to 77%. This research found for the first time that the lipid metabolism disorder caused by AFB1 and AFM1 was one of the main causes contributing to enterotoxicity, which could provide new insights into the toxic mechanisms of AFB1 and AFM1 in animals and humans.

## 1. Introduction

Aflatoxins are mainly secondary metabolites produced by *Aspergilus flavus* and *Aspergillus parasiticus* that induce toxic or carcinogenic effects. The main direct and indirect routes of human and animal exposure are ingestion of contaminated food or contact with environment aflatoxins [1]. Studies have shown that acute aflatoxin poisoning can cause abdominal pain, vomiting, and even death [2]. Another study by Akinrinmade et al. [3] pointed to AFB1-induced intestinal injuries in rats. Infants or children are negatively affected by aflatoxins, including nutrient maldigestion and malabsorption, chronic immune activation, impaired bone growth, and remodeling gastrointestinal diseases [4,5]. Among these toxins, Aflatoxin B1 (AFB1) is considered the most recurrent and also the most harmful. AFB1 often occurs in agricultural production and cereal crops, as well as in the soil ecosystem and environment [6,7]. A study of animal feed from China during 2018–2020 showed that AFB1 was present in 81.9% of the samples [8]. As a result of their stable physical and chemical properties, aflatoxins are currently recognized as some of the most toxic carcinogens. AFB1 has genotoxic, carcinogenic, teratogenic, and mutagenic activities [9,10,11,12]. Previous toxicological data indicated that AFB1 are group 1 human carcinogens [13]. AFB1 has been the research focus of most studies since its discovery, mainly because of the intermediate metabolite aflatoxin AFB1-exo-8,9 epoxide. As a highly unstable molecule, AFB1-exo-8,9 epoxide reacts with cellular macromolecules to induce metabolic disorders [14]. Another toxin causing great concern is Aflatoxin M1 (AFM1); after ingestion of AFB1 by cows, aflatoxin M1 (AFM1) remained in milk through hydroxylation (hence the M in the name) [1]. AFM1 was verified to have higher enteric carcinogenicity owning to its high polarity [15]. It is frequently found in human and animal milk [16,17] and infant formula [18,19]. Thus, it is possible for humans to come in contact with AFB1 and AFM1 through their daily dietary patterns of cereal and milk. Compared with single aflatoxins, combinations may cause more serious effects, posing a greater risk to animal and human health [20,21]. In particular, AFB1 and AFM1 are known to be closely associated with animal and human health, but the interaction mechanism is not clear, especially from a small molecular level.

The intestines are the first barrier to human contact with foodborne pollutants, and they play critical roles in maintaining human and animal health. The intestines have become one of the important targets of aflatoxins. AFB1 is absorbed rapidly from the small intestines into the mesenteric venous blood. However, many studies have focused on the mechanism of renal and liver toxicity [22,23,24], and there have been few studies of the combined enterotoxicity of AFB1 and AFM1. Our previous research revealed the additive and synergistic enterotoxic effects of a combination of AFB1 and AFM1 in vivo and in vitro [25,26]. Apart from our research, few studies have investigated the enterotoxicity of a combination of AFB1 and AFM1 and its underlying mechanisms, let alone from metabolite levels, especially for lipids metabolites to decode the mechanism. Intestinal metabolism is of considerable importance to health and disease [27,28]. Unlike other nutrients, the absorption way of lipids has its own distinguishing features [29]. The normal function of intestinal lipid metabolism is vital to ensure an adequate energy supply for the various organs in the body. The immune toxic effect of AFB1 mainly by reducing the intestinal production of metabolites, including short-chain fatty acids, deoxycholic bile acid, and indole [30]. 3,4,4’-trichloro-carbanilide exerted enterotoxicity mainly by disrupting intestinal bile acid and amino acid metabolism in mice [31]. In addition, a novel probiotic CP9 was proved to attenuate the enteric infection in IPEC-J2 cells induced by Escherichia coli through metabolic modulation [32]. Research of the enterotoxicity of AFB1 and AFM1 using metabolomics and lipidomics is needed. NCM460 is a normal human colonic mucosal epithelial cell line and has become an important in vitro experimental model for intestinal research owning to its potential characteristics before carcinogenesis when the culture is in vitro [33,34]. Hence, this study aimed to explore the enterotoxicity process of AFB1 and AFM1 in NCM460 cells at the metabolic level.

Metabolomics is a common method to reveal corresponding changes in metabolism derived from external stimuli [35]. It aims to monitor changes in metabolites so as to provide critical targets to study mechanisms at the metabolic level. Previous research has used metabolomics to explore the mechanisms of aflatoxicosis, with a focus on the kidneys and liver [36,37,38]. Lipidomics is a rapidly growing method derived from metabolomics, attempting to construct an all-embracing cellular liposome with lipid biology, technology, and medicine [39]. Recent research has shown that lipidomics can precisely identify markers of aflatoxin cytotoxicity [40]. Metabolomics and lipidomics are becoming emerging approaches for the mechanistic investigation of the targets of pollutants [41,42]. They have also been utilized in numerous research fields, including toxicology, drug development, and environmental science [41,43]. The relationship between polybrominated diphenyl ether-47 and breast cancer in mice was unveiled by using metabolomics and lipidomics [42]. These two omics methods were also performed in Neuro-2a cells to investigate the toxic effect of neonicotinoids imidacloprid and acetamiprid [41]. A combination of both omics methods could provide mechanistic clues about the biological processes of toxic substances, which could help us identify toxic targets more quickly. In addition, it can cover the majority of small molecules and thus offer a biomarker of cellular damage after toxicant exposure. So, it is important to study aflatoxins at a small molecular level.

However, no research has simultaneously revealed perturbations of AFB1 and AFM1 exposure on metabolites and lipids in cells. In the present study, we used metabolomics and lipidomics to investigate the changes in metabolites and lipids induced by AFB1 and AFM1 in NCM460 cells.

## 2. Results

### 2.1. Effect of AFB1 and AFM1 Treatment on NCM460 Cells Viability

The toxic effects of AFB1 and AFM1 (1.25, 2.5, 5, 10, 15 or 20 μM) on NCM460 cells were examined. Aflatoxins reduced cell viability in a dose−dependent manner (Figure 1). The half−inhibitory concentration (IC_50_) was calculated according to the dose−response of nonlinear regression. The IC_50_ of AFB1 was 7.287 μM, while that of AFM1 was 9.720 μM, whereas that of the combination IC_50_ was 5.813 μM. AFB1 induced stronger cytotoxicity than AFM1, and AFB1 + AFM1 was the most toxic for NCM460 cells. The IC_10_ and IC_20_ of AFB1 were 2.218 and 2.632 μM, respectively, compared with 2.193 and 2.675 μM for AFM1 based on the dose−response curve. According to the ISO standard (2009), a substance that reduces cell viability below 70% is regarded as cytotoxic after incubation with pre-cultured cells for at least 24 h. Among the concentrations above, 2.5 μM was the maximum for which the inhibition rate was no more than 30% for both AFB1 and AFM1. Thus, AFB1 and AFM1 alone and in combination at 2.5 μM were used for further metabolomics and lipidomics analysis.

### 2.2. Multivariate Analysis of NCM460 Cell Metabolic Profiles

Orthogonal projection to latent structure−discriminant analysis (OPLS−DA) was used to examine sample variance and distribution of metabolomics data within and between groups. With an orthogonal signal correction filter, OPLS−DA had greater discriminant ability compared with principal component analysis (PCA). It showed no significant deviation in all samples, therefore, no sample was discarded in subsequent analyses (Figure 2). The ellipse showed 95% of the distribution range, which indicated that the four groups of samples had no obvious outliers. Additionally, all aflatoxin groups were completely separated from the control group (Figure 2A−C), indicating the different expression profiles of metabolites between the control and treated cells. This implies that AFB1 and AFM1 induced metabolic disorders in NCM460 cells.

### 2.3. Differential Metabolites in NCM460 Cells Induced by AFB1 and AFM1

Differential metabolites were screened out by different pairwise comparisons, according to Dunn’s test (*p* < 0.05 and FC > 1.5). Figure 3A compared the results between the aflatoxin and control groups. Compared with the control group, AFB1 had 15 upregulated and 19 downregulated metabolites, while AFM1 had 14 upregulated and 18 downregulated metabolites. In the AFB1 + AFM1 group, there were 23 upregulated and 23 downregulated metabolites. Venn analysis was performed to find the specific and common target metabolites of single and combined aflatoxins. There were 10, 6, and 17 specific metabolites in the AFB1, AFM1, and AFB1 + AFM1 groups, respectively (Figure 3B). Boxplots depicted the level of these metabolites in the different groups (Appendix A). In addition, 19 common metabolites were found among the aflatoxin groups.

To distinguish the differences and similarities in the toxic effects of different aflatoxins, 52 significantly changed metabolites were used for heatmap analysis: 33 specific (10 for AFB1, 6 for AFM1, and 17 for AFB1 + AFM1) and 19 common metabolites. AFB1 affected the metabolites in orange font, such as choline, lysophosphatidylcholine (LysoPC) (16:1), LysoPC (18:1), and LysoPC (20:4) (Figure 3C). AFM1 affected the metabolites in pale blue/green font, including IMP and glutaric acid. AFB1 + AFM1 affected the metabolites in sky blue font, such as upregulated taurine and downregulated succinic acid. The metabolites in black font were commonly affected by AFB1, AFM1, and AFB1 + AFM1, including 11 downregulated and 8 upregulated metabolites.

### 2.4. Metabolic Pathway Analysis Affected by AFB1 and AFM1

Metabolite set enrichment analysis (MSEA) and metabolic pathway analysis (MetPA) were used to focus on the pathways and metabolite sets after aflatoxins treatment. Pathway analysis revealed that the metabolism of purine (*p* = 0.005) and glycerophospholipid (*p* = 0.014) were the pathways most significantly affected by AFB1 when compared with the control group (Figure 4A). AFB1+AFM1 treatment significantly affected the glycerophospholipid metabolism pathway of NCM460 cells (*p* = 0.043) (Figure 4B). For AFM1, there was no significantly affected pathway, but MSEA showed that AFM1 mainly affected fatty acid metabolites, such as propanoate, and β oxidation of very long−chain fatty acids (Appendix A). This may be because AFB1 and AFM1 exerted their toxicity in different ways. Glycerophospholipid metabolism was identified as the key pathway for AFB1 and AFB1 + AFM1, while AFM1 mainly disrupted fatty acid metabolism. AFB1 might be the predominant toxin when combined with AFM1. Thus, the different structural compositions of AFB1 and AFM1 may result in different enterotoxicity for NCM460 cells.

To explore further the different toxic effects of AFB1 and AFM1, KEGG analysis (https://www.omicshare.com/tools/, 21 July 2022) of significantly changing specific metabolites in different groups was carried out. Glycerophospholipid metabolism was the main pathway affected by AFB1 (Figure 5A). AFM1 induced changes in fatty acid degradation (Figure 5B), and AFB1 + AFM1 affected propanoate and sulfur metabolism (Figure 5C). Glycerophospholipid metabolism was affected by AFB1, while AFM1 was inclined to affect fatty acid degradation. AFB1 + AFM1 interfered with more metabolic pathways, and the most affected was propanoate metabolism. Based on the results of metabolomic pathway analysis, aflatoxins mainly affect lipid metabolism. To elucidate further the specific changes in lipids metabolism, lipidomics analysis was performed.

### 2.5. Lipid Profiles on NCM460 Induced by AFB1 and AFM1

To assess the changes in lipid metabolites, lipid profiles in NCM460 cells were determined after aflatoxin treatment. In total, 27 lipid subclasses were detected, including 441 lipid molecules. Triacylglycerol (TAG) was the most frequently detected, comprising 128 lipid molecules and accounting for 29% of all lipids, followed by cardiolipin (CL), diacylglycerol (DAG), and sphingomyelin (SM) (Figure 6A). PCA of lipids demonstrated clear separation among the three treatment groups and control group, which suggested significant differences in the effects of aflatoxins on lipid metabolism in NCM460 cells (Figure 6B). This also proved that the metabolic effect of AFB1 and AFM1 was mainly through changes in lipid metabolism.

### 2.6. Analysis of Differential Lipids in NCM460 Cells

According to the Dunn’s test *p* < 0.05 and FC > 1.5, differentially expressed lipids were filtrated by pairwise comparison of control and aflatoxin−treated groups. Compared with the control group, there were 63 (22 upregulated and 41 downregulated), 23 (6 upregulated and 17 downregulated), and 52 (40 upregulated and 12 downregulated) lipids that satisfied the screening conditions in the AFB1, AFM1 and AFB1 + AFM1 groups (Figure 7A). The changes in lipid metabolism on NCM460 cells induced by AFB1 were more intense than those induced by AFM1. Venn analysis was performed to find the specific differentially expressed lipids between the cells treated with AFB1 and AFM1 alone and in combination. There were only five lipids identified in common among the AFB1, AFM1 and AFB1 + AFM1 groups, which showed that AFB1 and AFM1 had different effects on lipid profile (Figure 7B). There were 34, 11, and 30 specific lipids identified in the AFB1, AFM1 and AFB1 + AFM1 groups, respectively.

Heatmap analysis was used to display the expression profile of the specific lipids identified above. For AFB1, 34 specific lipids were identified, 27 lipids were decreased, and 12 of these, including CL, TAG, phosphatidyl ethanolamine (PE) and SM, accounted for 63% of the total downregulated lipids (Figure 7C). Only seven metabolites were increased by AFB1: TAG46:2 (18:2), TAG54:6 (18:2), TAG60:3 (18:2), TAG 58:3 (24:0), lysophosphatidic acid (LBPA)38:6 and LBPA36:4, and phosphatidylcholine (PC) 36:4. AFM1 tended to exert its toxicity effects through reducing free fatty acids (FFAs), phosphatidylinositol, CL, cholesteryl and phosphatidylglycerol (PG) (Figure 7D), and CL and PG accounted for 70% of the total. Only the level of TAG48:0 (16:0) was increased by AFM1. AFB1 + AFM1 significantly increased all the TAGs (Figure 7E), and PC40:7, DAG32:0 (16:0/16:0), PC36:3p, LysoPC20:4 and PE42:2p were decreased. In short, AFB1 exerted its toxic effects on NCM460 cells through changing the level of CL, TAG, PE, and SM; AFM1 affected CL and PG; while increased metabolic level of TAG played a critical role in AFB1+AFM1−induced enterotoxicity.

## 3. Discussion

Aflatoxins are regarded as a health issue of increasing international concern, and information on potential ways to mitigate the negative effects of AFB1 and AFM1 exposure remains limited. Ammoniation and adsorption on clays or organic adsorbents are commonly used in feed [44,45]. Recently, studies have paid attention to the AFM1 binding ability of different strains of Lactobacilli around AFM1 in milk [46,47]. The toxicity of AFB1 and AFM1 has been determined in various organs and tissues in vivo and in vitro, and the intestines are key target organs. Aflatoxins have been shown to impair intestinal mRNA and protein levels [26,48], but few studies focused on metabolism and lipid levels. Therefore, the present study investigated the metabolic effects and underlying toxic mechanisms of AFB1 and AFM1 on NCM460 cells, which could provide a theoretical basis for the risk assessment and treatment of enterotoxicity.

The viability of NCM460 cells was reduced by AFB1 and AFM1 in a dose-dependent manner. The order of severity of toxic effects on NCM460 cells was AFB1 + AFM1 > AFB1 > AFM1, as evaluated by IC_50_ (Figure 1). Only one study to assess IC50 value on the intestinal cell was found, pointing out that the IC50 of AFB1 on Caco-2 cells was 39.1 ± 13.2 μM, while AFB1 had so few cytotoxic effects on the two HCT116 cell lines that IC50 could not be calculated [49]. The IC50 value on the Caco-2 cell was higher than our results, which could be explained by the different cell lines. The cancer cell line Caco-2 tolerance of aflatoxins is stronger than normal intestinal cell line NCM460. This article also confirmed that AFB1 has different toxicity to different intestinal cells, and there is also a great difference in IC50 value. Regardless of the number of specific differential metabolites and enrichment pathways, the effects of AFB1 + AFM1 were stronger than those of AFB1 and AFM1 alone, which was consistent with the phenotypic tests.

Metabolomics analysis pointed out that glycerophospholipid was significantly increased by AFB1. Glycerophospholipids are an essential part of the lipid bilayer and have an important influence on membrane properties and dynamics [50,51]. It also alleviated ulcerative colitis in mice [52]. Thus, the significant changes in glycerophospholipids induced by AFB1 may influence the overall metabolism of NCM460 cells, which infers that AFB1 causes intestinal lesions and even cancer by disrupting glycerophospholipid metabolism [37]. The decreased level of LysoPC (16:1), LysoPC (18:1), and LysoPC (20:4) and increased choline in the AFB1 treatment group indicates that AFB1 can damage the intestinal barrier and immune function, which may even develop into cancer [53,54]. Some studies confirmed that AFB1 exerted a negative effect on intestinal cancer cells [55]. In addition, both specific metabolites LysoPC and xanthosine participated in intestinal inflammation in the AFB1 group [56,57]. Changes in the level of these metabolites are also reported to be related to intestinal inflammation and mucosal homeostasis [58]. Xanthosine was also found in the liver [59], but the opposite trend was presented after AFB1 treatment. Higher levels of choline induced cell swelling because choline, phosphocholine, and PC are essential elements for the structural integrity of cell membranes [60]. Choline was increased by AFB1 in this study, which has become the endogenous biomarker of cancer [53]. Thus, AFB1 exerted enteric toxicity through disruption of intestinal homeostasis, reflected by the metabolic disorder of glycerophospholipids. The specific influence of AFM1 was reflected in fatty acid degradation, and glutaric acid was significantly enrichment in this pathway (Figure 5B). Recent research identified that glutaric acid may be involved in regulating intestinal microbiota [61], with increased glutaric acid levels exerting a negative effect on the intestine. Therefore, the influence of fatty acid metabolism may be the main cause of AFM1-induced enterotoxicity. Propanoate metabolism and sulfur metabolism were specifically enriched in the AFB1 + AFM1 group (Figure 5C), and the downregulation of succinic acid and upregulation of taurine were identified in these two pathways. Prior research has demonstrated that succinic acid plays a critical role in regulating intestinal immune responses to external stimulation [62,63]. AFB1 + AFM1 increased the taurine level, which may induce intestinal proinflammatory and oxidative effects [64,65]. Taurine regulates the production of short-chain fatty acids and the occurrence of colitis [66,67]. Therefore, increased taurine probably promotes intestinal inflammatory response. The effects on glycerophospholipid metabolism, fatty acid degradation, and propanoate metabolism all reflect that AFB1 and AFM1 affect lipid metabolism.

Lipid metabolites were identified as the pivotal type of differential metabolites; thus, it is necessary to define the role of lipid metabolism in intestinal injury induced by AFB1 and AFM1. Aflatoxins are highly lipophilic [68], and lipid metabolic disorders are a common hallmark of aflatoxicosis [59,69]. Each part of the intestine has its own functions and maintains normal metabolism of lipids. Dietary lipids undergo digestion, emulsification, hydrolysis, and micellization in the duodenum. The jejunum promotes lipid assimilation, and the ileum provides the place for this process and enterohepatic circulation [70]. Disorders of lipid metabolism occur when aflatoxins decrease the length of the small intestine induced [71]. A lipidomics study identified 48 lipids among the DAGs, TAGs, FFAs, and phospholipids that have metabolic dysregulation upon exposure to mycotoxin [72]. To understand further the mechanism of aflatoxin-induced enteric toxicity at the metabolic level, we performed lipidomics analysis.

PCA of lipidomics analysis proved the different focus of AFB1 and AFM1 on lipid metabolism (Figure 6B). In addition, the results showed that AFB1 evoked greater changes than AFM1 in lipid metabolism. AFB1 exerted enterotoxic effects on NCM460 cells mainly by changing the levels of CL, TAG, PE, and SM (Figure 7C). The changed level of CL was reported to affect mitochondrial function and inflammatory response [73]. The synthesis of SM participated in cell normal physiology, and dysregulation of SM level may affect the tumor transformation [74]. PC and DAG were also found in the AFB1 group. Synthesis of PC is necessary for lipid absorption in dietary and metabolic homeostasis in the intestines [75]. Fecal lipidomics in F344 rats revealed that AFB1 induced intestinal epithelial toxicity through changes in the level of PE and DAG [55]. The decrease in PC32:1, PE, and DAG may be accompanied by inflammation. Previous research showed that AFB1 disrupted intestinal homeostasis, which was marked by a reduction in TAG and cholesterol [76]. Another study pointed out that the plasma and liver levels of TAG in rats were increased after acute exposure to AFB1 [69]. The reason for the inconsistent results may be that there are many different types of TAG. In the present study, AFB1 simultaneously increased and decreased the different TAG molecules. The specific mechanism of changes at different molecular levels is worth further investigation.

The enterotoxic effects of AFM1 were mainly mediated through decreasing levels of CL and PG (Figure 7D). Different CL species have been identified in cells treated with AFB1 or AFM1, which can block Toll-like receptor 4 response to lipopolysaccharides and maintain intestinal homeostasis [77]. In contrast, decreased CL level may induce intestinal homeostasis disorder. PG was reported to reduce the production of tumor necrosis factor-α and exert anti-inflammatory effects in vitro [78]. The decreased level of PG in our study indicated that AFM1 induced a proinflammatory response in NCM460 cells.

The accumulation of TAG accompanied by inhibition of intestinal fat absorption causes severe adiposity [79]. Ugbaja et al. (2020) reported that the synthesis of TAG occurred via complex pathways that involved PC or DAG or both membrane lipids [80]. Thus, the increase in TAG by AFB1 + AFM1 meant that not only were cellular lipids altered, but also priority interfered with metabolic pathways of lipids even involved in other cellular. Therefore, AFB1 + AFM1 exerted enterotoxicity by raising the level of TAG metabolism. Targeting lipid metabolism or signaling is becoming a trend for disease therapeutics [81]. Regulation of lipid metabolism alters the antitumor function of T cells in cancer mouse models [82,83]. The regulation of lipid metabolic pathways can alter the production and function of Treg cells, which may also be applied to the treatment of autoimmune diseases [84].

How AFB1 and AFM1 altered the level of lipids in NCM460 cells remains unclear. We propose a hypothesis that AFB1 and AFM1 exert their enterotoxicity through lipid dysmetabolism. This needs to be confirmed in future studies.

## 4. Conclusions

Enterotoxic effects of AFB1 and AFM1 alone and in combination were investigated by simultaneous metabolomics and lipidomics analysis. Metabolomics analysis revealed AFB1 and AFM1, both individually and in combination, disrupt lipid metabolism in different specific pathways: glycerophospholipid metabolism was the dominant interrupted by AFB1; fatty acid degradation was disrupted by AFM1; propanoate metabolism was the main interrupted in AFB1 + AFB1. A combination of AFB1 and AFM1 caused more extensive metabolic disturbances than either aflatoxin alone. AFB1 and AFM1 played an enterotoxic role by disrupting intestinal lipid metabolism. Further, lipidomics analysis revealed that 75 specific lipids with significant changes were screened after AFB1 and AFM1 treatment. Of all the differential lipids, TAGs were the most frequently observed, with a proportion of 43%. Importantly, AFB1 caused major perturbations in lipid metabolism levels in NCM460 cells and might show greater enterotoxicity compared with AFM1. The results pointed the lipid metabolism disorder caused by AFB1 and AFM1 should be emphasized in environmental toxicological studies, which provide a novel direction for the study of aflatoxin-related health issues, as well as novel molecular targets for preventing AFB1 and AFM1 contamination. Lipid metabolism identified in this study also can be a regulatory mechanism for reducing the damage of aflatoxins and may even be for clinical disease treatment.

## 5. Materials and Methods

### 5.1. Chemicals and Reagents

AFB1 and AFM1 (95% purity) were provided by Pribolab (Qingdao, Shangdong, China), and a stock liquor of 200 μM was formulated by dissolving in dimethyl sulfoxide (DMSO) (Sigma−Aldrich, St. Louis, MO, USA). RPMI 1640 and fetal bovine serum (FBS) used for cell growth were from Gibco (Grand Island, NY, USA). The antibiotics (100 U/mL penicillin and 100 μg/mL streptomycin) and nonessential amino acids (NEAAs) were obtained from Life Technologies (Carlsbad, CA, USA). L-Glutamine was supplied by Sigma-Aldrich. 3-(4,5-dimethylthiazol-2-yl)-2,5-diphenyl-2H-tetrazolium bromide (MTT) and phosphate buffer solution were provided by Beyotime Biotechnology (Shanghai, China).

### 5.2. Cell Culture and Treatment

The normal human colon cell line NCM460 cells were purchased from the American Type Culture Collection (Manassas, VA, USA) and cultured as previously [85]. NCM460 cells were maintained in RPMI160 with 10% (*v*/*v*) FBS, 1% antibiotics, L–glutamine, and NEAA in a 37 °C incubator in 5% CO_2_ (Thermo, Waltham, MA, USA). Water used in the incubation system was purified through a Milli−Q system (Millipore, Bedford, MA, USA). When NCM460 cells were seeded in 6-well and 96−well plates (Corning, Corning, NY, USA) at 2 × 10^5^ cells/well and 5 × 10^4^ cells/well for 24 h and challenged with AFB1/AFM1 (0−20 μM) for 48 h. The cells in 96−well and 6−well were respectively collected for subsequent cell viability, metabolomics, and lipidomics analysis.

### 5.3. Cell Viability Assay

The viability of NCM460 cells in 96−well plates after exposure to AFB1 and AFM1 was evaluated by using the MTT assay. After the cells grew up to a confluence of 90%, AFB1 and AFM1 were added to 96−well plates (5 × 10^4^ cells/well) at 0, 1.25, 2.5, 5, 10, 15, and 20 μM for 48 h. A total of 5 mg/mL MTT solution was prepared by dissolving 25 mg MTT with 5 mL MTT solvent. The cells were treated with 10 μL 5 mg/mL MTT solution incubated for 4 h. Then the cells were cultured with 100 μL formazan solvent (37 °C for 3−4 h). The plate was gently shaken for 10 min to completely dissolve the crystals, and absorbance was recorded at 570 on the microplate reader (Thermo Scientific, Waltham, MA, USA). GraphPad Prism 8 was used to analyze the IC10, IC20, and IC50 according to the dose–response of nonlinear regression.

### 5.4. Metabolomics Analysis

Metabolome extraction was performed as described previously [86]. Each treatment group had eight duplicate samples. Cell samples were mixed with 80% ice−cold methanol in water, incubated at 1500 rpm at 4 °C for 30 min, and centrifuged at 12,000 rpm at 4 °C for 10 min. The supernatant obtained above was transferred into a clean 1.5 mL centrifuge tube. Genevac miVac (Tegent Scientific, Shanghai, China) was used to redissolve the dried extracts in water with 5% acetonitrile. Moreover, the upper liquids were collected for LC−MS analysis.

The LC−MS parameters were as described previously [86]. ACQUITY UPLC HSS T3 1.8 μm, 3.0 × 100 mm columns were used for chromatographic analysis. LC−MS experiments of Q−Exactive MS with a HESI probe controlled by Xcalibur 2.3 software were carried out on the Dionex UltiMate3000 HPLC system. The MS detection parameters were sheath gas flow rate 40 psi, auxiliary gas flow rate 11 arb, sweep gas flow rate 0, and full MS scan, and the scan range was 80−1000 *m*/*z*, capillary temperature 350 °C, spray voltage 3.5 kV and 3.2 kV, respectively, for positive and negative mode, s-lens RF level 55, and auxiliary gas heater temperature 220 °C. The Q−Orbitrap performed data−dependent scans in full MS/dd−MS2 to obtain product ion information with normalized collision energy (NCE), and the mass resolution was set at 17,500 FWHM (*m*/*z* 200) and NCE 35%. All detected ions were compared with HMDB, METLIN, and standard references to annotate ion ID. High−purity nitrogen was used as the atomization gas and the collision gas for higher energy collisional dissociation. Differential metabolites were screened based on the Dunn test *p* < 0.05 and fold change (FC) >1.5. Two methods were adopted for pathway analysis: metabolite set enrichment analysis (MSEA) and metabolic pathway analysis (MetPA). MSEA analysis can find out the set of metabolites that are enriched in this group of metabolites. The test method used is Over−representation analysis (ORA). MetPA has two important components: ORA and pathway topological analysis. ORA detects whether a metabolite set includes more metabolites with significant differences than would be expected at random. ORA tests can be either a hypergeometric test or Fisher’s exact test.

### 5.5. Lipidomic Analysis

An improved version of Bligh and Dyer’s method was used to extract lipids [87]. Each group was composed of three duplicate samples, and a total of four groups were applied in the lipidomics study. NCM460 cells were homogenized in 750 µL chloroform: methanol: MilliQ H_2_O (3:6:1) (*v*/*v*/*v*) and incubated at 1500 rpm at 4 °C for 1 h. After incubation, chloroform (250 µL) and deionized water (350 µL) were used to induce phase separation. The lower organic phase containing lipids was extracted into a clean tube after centrifugation. Chloroform (450 µL) was added to the remaining cells in the aqueous phase to repeatedly extract lipids, and the lipid extracts were pooled into one tube and dried in the OH mode of the SpeedVac. The samples were kept at −80 °C for further testing.

An ExionLC−AD combined with Sciex QTRAP 6500 PLUS was used to perform lipidomic analyses at LipidALL Technologies (Guangzhou, China) [88]. Each lipid of polar was separated by normal-phase HPLC using a TUP−HB silica column (i.d. 150 × 2.1 mm, 3 µm) under the specific conditions: mobile phase A (chloroform: methanol: ammonium hydroxide, 89.5:10:0.5) and mobile phase B (chloroform: methanol: ammonium hydroxide: water, 55:39:0.5:5.5). Multiple reaction monitoring (MRM) transitions were established to comparative analysis of various polar lipids. Individual lipid species were quantified by referencing spiked internal standards. Free fatty acids (FFA) were quantitated using FFA 19:0 (Cayman Chemicals). Glycerol lipids, including diacylglycerols (DAG) and triacylglycerols (TAG), were quantified using a modified version of reverse−phase HPLC/MRM. Separation of neutral lipids was achieved on a Phenomenex Kinetex−C18 column (i.d. 4.6 × 100 mm, 2.6 µm) using an isocratic mobile phase containing chloroform: methanol:0.1 M ammonium acetate 100:100:4 (*v*/*v*/*v*) at a flow rate of 300 µL for 10 min. Levels of TAG were calculated by referencing the spiked internal standard TAG (16:0) 3-d5 obtained from CDN isotopes, respectively. DAG was quantified using d5-DAG16:0/16:0 and d5-DAG18:1/18:1 as internal standards (Avanti Polar Lipids). Free cholesterols and cholesteryl esters (CEs) were analyzed under atmospheric pressure chemical ionization mode on an Agilent 1260 HPLC connected to Sciex QTTRAP 5500, using d6-cholesterol and d6-C18:0 CEs (CDN isotopes) as internal standards [89].

Principal component analysis (PCA) was used to reduce dimensions that identify a small new set of independent variables that best represent the information of the original data so that the transformed data were easier to visualize and interpret. FactoMineR package was used for PCA. Plots of differential metabolites, boxplots, and barplot are all based on hypothetical tests. We used the ggplot2 package for these plots. Original concentrations of individual lipids (MFP) were used in Dunn’s test. Concentrations of individual lipids (MFP) were standardized to z scores for the heatmap plot. All statistical analyses were performed using R 4.0 software. Two-sided *p* < 0.05 was considered statistically significant. Dunn’s *t*-test with *p* < 0.05 and FC > 1.5 were identified as differential lipids.

### 5.6. Statistical Analysis

Statistical analysis of changes in cell viability and inhibition after AFB1 and AFM1 treatment compared with the control cells was performed by GraphPad Prism 8.0 (San Diego, CA, USA). Analysis of variance combined with Tukey’s multiple analysis was used to analyze the difference after confirming normality by Shapiro−Wilk tests. *p* < 0.05 represented statistically significant differences.

## Figures and Tables

**Figure 1 toxins-15-00255-f001:**
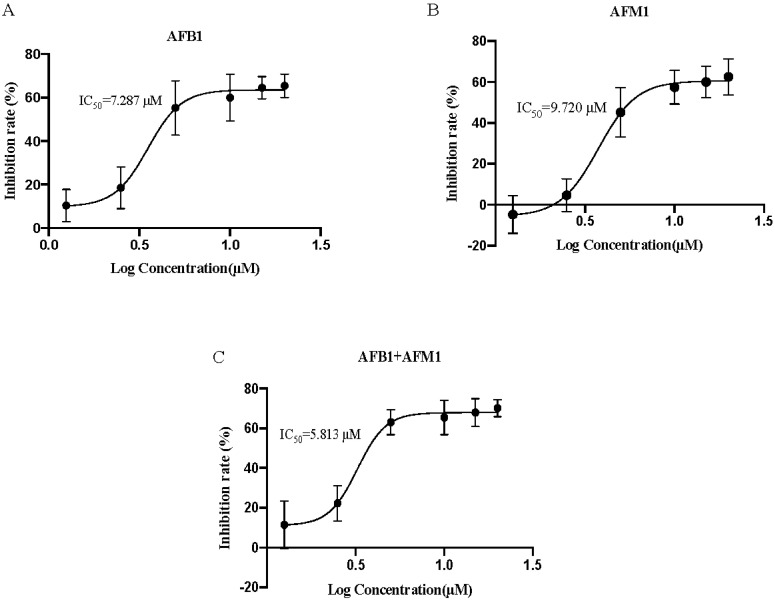
Dose−response curves of cell viability for effect of aflatoxin B1 (AFB1) (**A**), aflatoxin M1 AFM1 (**B**), and combination of AFB1 and AFM1 (AFB1+AFM1) (**C**) on NCM460 cells (*n* = 10).

**Figure 2 toxins-15-00255-f002:**
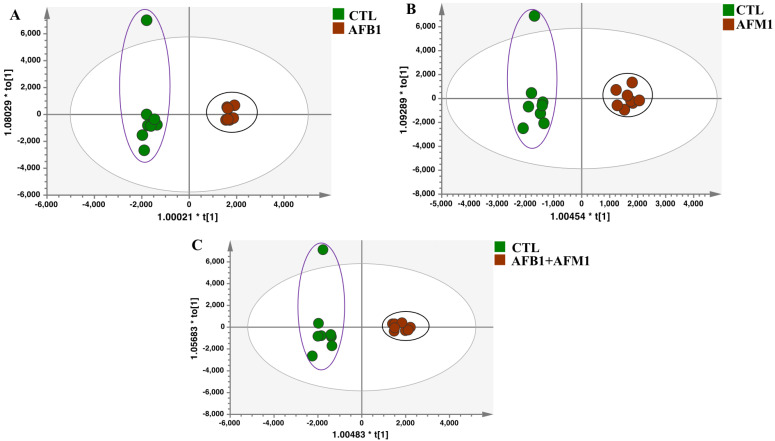
Overview of metabolic profiles on NCM460 cells after aflatoxin treatment. Orthogonal projection to latent structure−discriminant analysis (OPLS−DA) for different treatment groups (*n* = 8): (**A**) AFB1, (**B**) AFM1, and (**C**) AFB1 + AFM1 compared with control (CTL).

**Figure 3 toxins-15-00255-f003:**
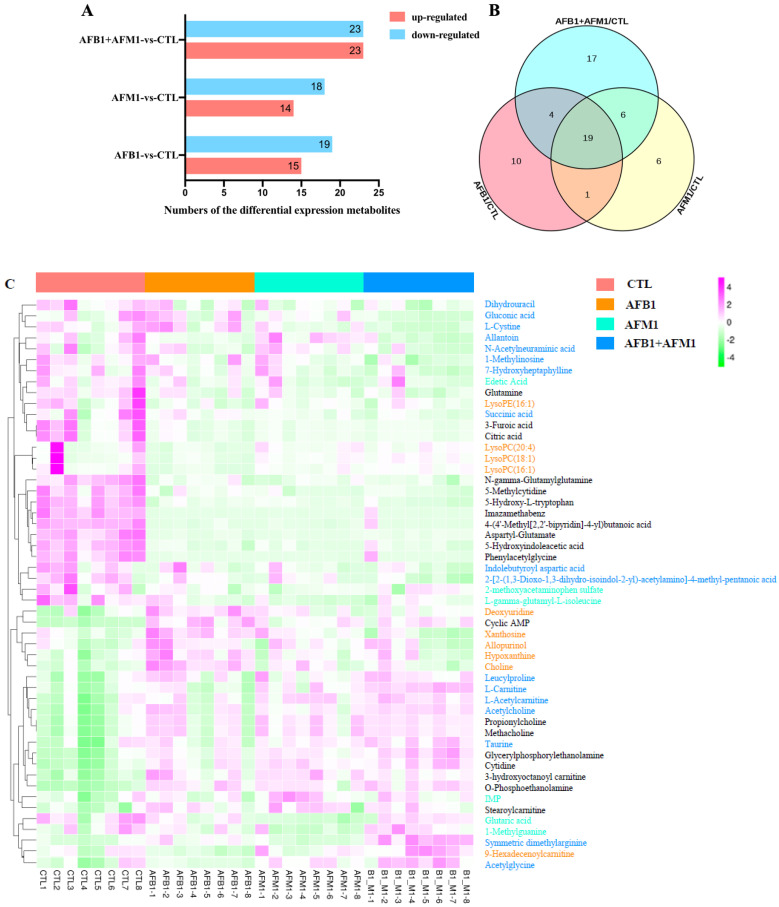
The analysis of differential metabolites after aflatoxins treatment alone or in combination. (**A**) The number of differential metabolites. (**B**) The Venn analysis of different metabolites. (**C**) Heatmap visualization displaying a relative abundance of specific and common metabolites (*n* = 8). A total of 52 significantly changed specific common metabolites are listed on the map. Each row in the map represents one metabolite, and each column represents one sample. The gradient color bar from purple to green denotes the concentrations of metabolites from high to low. The metabolites with different colors on the right represent the metabolites of different treatment groups, in which orange is the specific metabolite of the AFB1 group, pale blue-green is AFM1, sky blue is AFB1 + AFM1, and black is the common differential metabolites.

**Figure 4 toxins-15-00255-f004:**
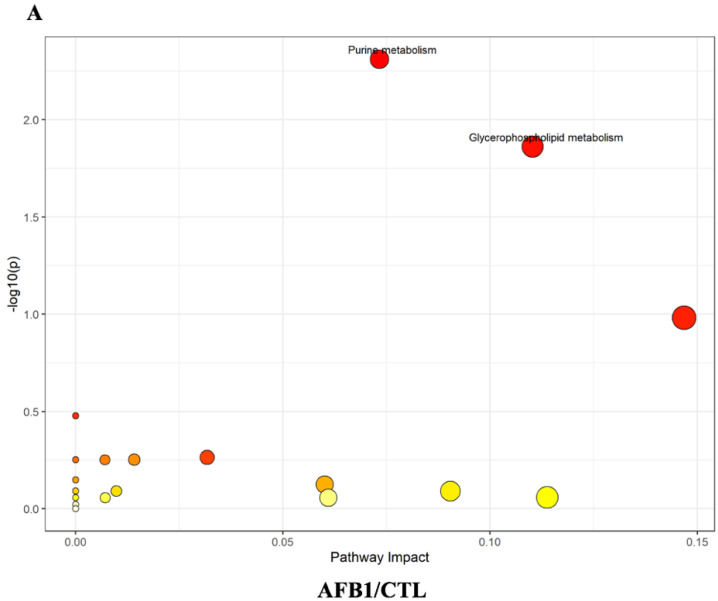
Metabolic pathway analysis (MetPA) of differentially expressed metabolites on NCM460 after aflatoxins treatment, (**A**) AFB1 and (**B**) combination of AFB1 + AFM1. The darker the color, the larger the circle, the higher the degree of enrichment.

**Figure 5 toxins-15-00255-f005:**
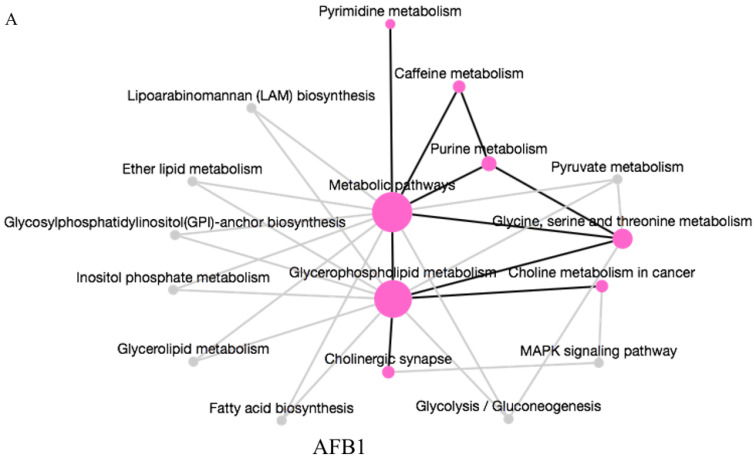
KEGG pathway enrichment analysis of specific metabolites in (**A**) AFB1, (**B**) AFM1, and (**C**) combination of AFB1 + AFM1. The size of the node indicates the degree of enrichment. The solid line indicates that there is a connection relationship between pathways, and the dotted line indicates the connection line between the complementary pathway and the input pathway.

**Figure 6 toxins-15-00255-f006:**
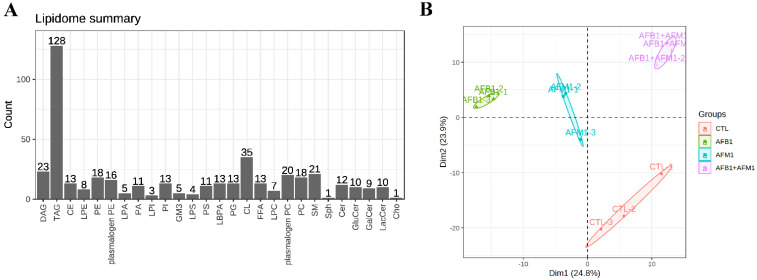
Lipids profiles on NCM460 cells after single and combination exposure of AFB1 and AFM1. (**A**) Subclasses and quantity of detected lipids. (**B**) PCA analysis of lipids, each point in the quadrants represents a single sample (*n* = 3).

**Figure 7 toxins-15-00255-f007:**
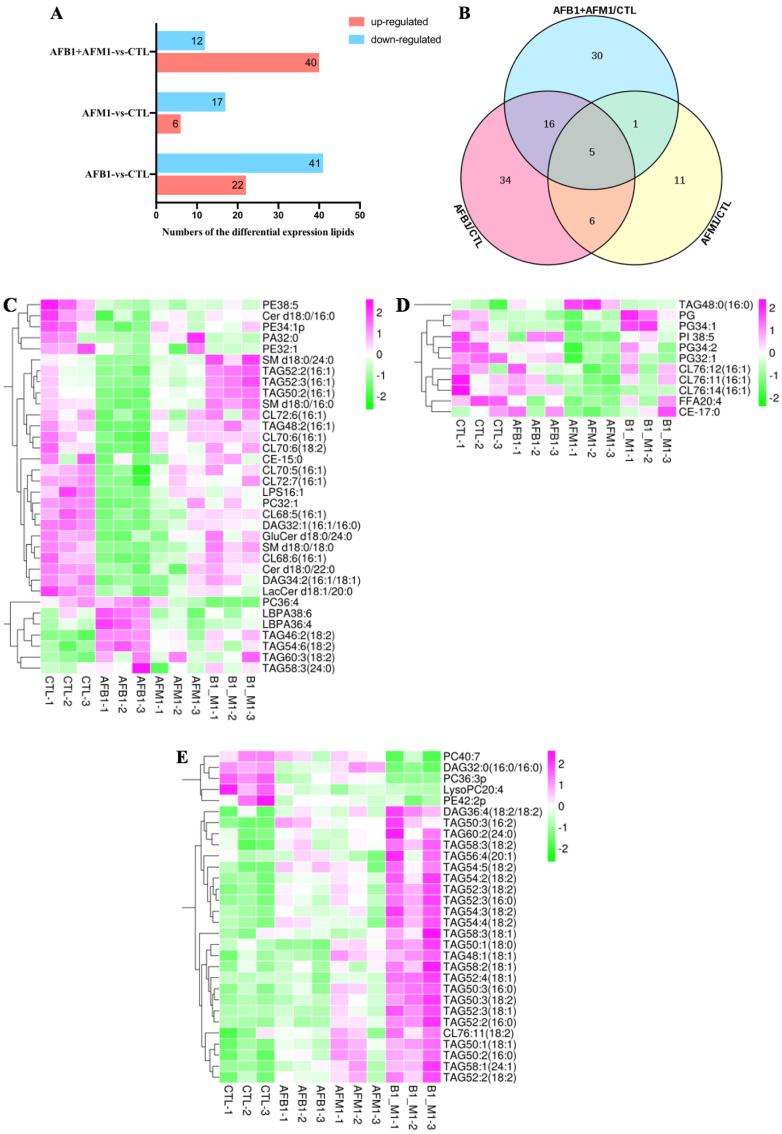
The analysis of differential lipids induced by single and combination of aflatoxins. (**A**) The numbers of the differential lipids after pairwise comparisons. (**B**) Venn analysis of the differential lipids compared with the control group (CTL). The heatmap analysis of specific metabolites in (**C**) AFB1, (**D**) AFM1, (**E**) AFB1 + AFM1 (*n* = 3).

## Data Availability

Data are available upon request; please contact the contributing authors.

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
