# Peer review of "Integrated Metabolomics and Lipidomics Analysis Reveals Lipid Metabolic Disorder in NCM460 Cells Caused by Aflatoxin B1 and Aflatoxin M1 Alone and in Combination"

_toxins, 2023, doi:10.3390/toxins15040255_

Round 1

Reviewer 1 Report

The article clearly illustrates the effects of aflatoxin B1 and M1 on the human cell line NCM460.

The conclusions are supported by experiments.

however, It is necessary to modify and/or clarify some points before publication:

line 95_96: needless to indicate the statistical system used in the results, it has already been indicated in the materials and method.

the caption of the figures is not exhaustive. Indicate the acronym in full. How many types of experiments (duplicate, triple?)

In all figures, the graphs do not indicate the significance well, not even in the results. Please improve them

line 35-36: the toxicity of Aflatoxin has been investigated also by other authors. Please add the following publication doi: 10.3389/fvets.2021.822227.

Reviewer 2 Report

The article explores the effects of aflatoxin B1 (AFB1) and aflatoxin M1 (AFM1) on lipid metabolism in NCM460 cells using comprehensive metabolomics and lipidomics analyses. The study shows that the combination of AFB1 and AFM1 induced more metabolic disturbances than either aflatoxin alone. The dominant pathways that were interfered with by AFB1, AFM1, and AFB1+AFM1 were glycerophospholipid metabolism, fatty acid degradation, and propanoate metabolism. The study concludes that lipid metabolism disorder caused by AFB1 and AFM1 is one of the main contributors to enterotoxicity. The study has the potential to advance thinking in the field, however, a few comments need to be considered to further hone the manuscript.

Major comments:

It would be helpful to describe how AFB1 and AFM1 were selected as the focus of the study and why NCM460 cells were chosen as the experimental model.

The article lacks clarity regarding the methods used for the metabolomics and lipidomics analyses. Specifically, more information on the data analysis and methods is needed.

The article should include more information on the specific metabolites and lipids that were affected by AFB1 and AFM1. Perhaps additional assays would help validate the enzymatic activities of the major pathways involved. It is not very clear how KEGG pathway analysis is inferred.

The authors should describe the significance of the identified pathways in the context of AFB1 and AFM1 toxicity.

It would be helpful to include information on potential methods for mitigating the negative effects of AFB1 and AFM1 exposure.

Minor comments:

The abstract should include more information on the methods used in the study.

The introduction should include more information on the importance of studying aflatoxins at the molecular level.

The authors should clarify how they obtained their results and how they can be used in the future.

The article could benefit from a more detailed explanation of the relationship between aflatoxins and cancer.

The article should include more information on the health effects of AFB1 and AFM1 exposure in animals and humans.

The authors should provide more detail on the specific metabolites and lipids that were analyzed.

The discussion section should be expanded to include more information on the clinical relevance of the findings.

Reviewer 3 Report

Dear Editor, Dear Authors,

I evaluated the manuscript « Integrated metabolomics and lipidomics analysis reveals lipid metabolic disorder in NCM460 cells caused by aflatoxin B1 and aflatoxin M1 alone and in combination »

In this study, the authors used omic approaches to study the effect of Aflatoxin B1 (AFB1) and aflatoxin M1 (AFM1) on NCM460 cell using metabolomic and lipidomic. The authors exposed the cells to AFB1, AFM1 and cocktail of them. Combination of AFB1 and AFM1 induced more extensive metabolic effect, with AFB1 having stronger effect than AFM1. More precisely, the data show effects on glycerophospholipid, fatty acid degradation, and propanoate metabolism. Those results suggest that attention should be paid to lipid metabolism after AFB1 and AFM1 exposure. The authors identified numerous lipids specifically altered such as cardiolipin (CL) and triacylglycerol (TAG). Conclusions of the authors were that AFB1/AFM1 induce alterations of lipid metabolism that could play a role in toxicity to enterocytes.

I found the study interesting and manuscript clear.

Please find below my comments and suggestions :

Major :

AFM1 is classically found much less toxic than AFB1. Here the IC50 of the two are quite close. Is it expected ? Can the authors comment on that in their discussions, may be comparing with data obtained with other intestinal cells models such as Caco-2 or other cell type ?

AFB1 is known to cause severe lipids and metabolic alterations in the liver. Can the authors compare their data with the existing ones on liver ? Are the same lipids affected ? Are the same lipids up-regulated and downregulated in intestine and liver ?

From a physiological point of view, why affecting lipid metabolism is important for AF1 effect on the gut rather than the adipocytes or liver cells ?

Minor :

« The supernatants were treated with 100 μL 0.5 mg/mL MTT solution incubated for 4 h. Then the cells were cultured with 100 μL formazan solution to dissolve methylazan crystals after discarding MTT solution » : please correct this section of the Materials & Methods as the MTT is not added to the supernatant but to the cells and that the cells can’t be cultured with formazan solution… Formazan is inside the cells and is released when treating the cells with DMSO.

regards

Reviewer 4 Report

The manuscript presents little exploited points of AFB1 and AFM1 toxicity which are the kind of cell and its lipidomic properties. However, the method description needs to be improved, specifically the data treatments and the choice of the statistic tests.  There are many abbreviations in the manuscript which make it difficult to understand the comments and meaning of the results. I suggest a list of the acronyms meaning. Supplementary material with statistical pieces of information would be interesting. 

Round 2

Reviewer 1 Report

It is ready for publication, thanks